# Drug use among young women in Pakistan: A qualitative study of socioeconomic and psychological perspectives

Muhammad Suhail Khan[1] ⬤, Yao Dewei[1], Wu Zongyou[1], Aman Khan[2], Ibrahim[3] and Anastasiia Pavlova[4]

[1]Department of Sociology, School of Sociology and Political Science, Anhui University, Hefei, P.R. China; [2]Department of Sociology, School of Public Administration, Hohai University, Nanjing, P.R. China; [3]Department of Sociology, University of Malakand, Khyber Pakhtunkhwa, Pakistan and [4]Department of International Trade, School of Economics, Anhui University, Hefei, P.R. China

## Research Article

**Keywords:**
drug use among young women; peer influence; mental health; socioeconomic factors; Pakistan

**Corresponding author:**
Muhammad Suhail Khan;
Email: sohail.khattak1@yahoo.com

## Abstract

Drug use among young women has severe consequences for their mental health, increases their developmental vulnerability and highlights the global problem of drug addiction. The purpose of this study was to investigate the socioeconomic and psychological factors influencing drug use among young women in Khyber Pakhtunkhwa, Pakistan. The study used a qualitative research design. We collected data from 12 women aged 18–21 years via in-depth qualitative interviews conducted in Mardan and Peshawar from March to June 2022. Research shows that young women frequently use drugs due to peer pressure, emotional challenges and financial concerns, which significantly impact their lives. The study emphasizes the value of cultural intervention programs for young women, concentrating on the region's mental health services, economic empowerment and gender-specific peer support networks.

## Impact statement

In this study, we examine the drug use among young women and their socioeconomic status, psychological characteristics and cultural influences in the Khyber Pakhtunkhwa province of Pakistan. The call for culturally sensitive and gender-sensitive approaches lays the foundation for the development of targeted treatment and public health programs. The results point to the need for comprehensive mental health care, community outreach and educational programs targeting youth in patriarchal societies. This approach challenges the stigma surrounding drug use and contributes to a more profound understanding of drug dependence as a multifaceted public health problem. This study contributes to global debates on gender, drug policy and health equity, and it suggests that similar strategies should be applied in other conservative contexts.

## Introduction

Drug use and addiction represent significant public health issues, with extensive consequences for people, families and communities. The World Health Organization states that drug use results in millions of deaths each year, facilitates the transmission of infectious diseases such as human immunodeficiency virus and hepatitis C, and aggravates chronic health conditions (Lemahieu and Me, 2014; Lanini et al., 2016; World Health Organization, 2018; Scheibe et al., 2020; Verma et al., 2020). Women are more prone to relapse, withdrawal symptoms and negative social consequences than men (Sun, 2007; Catalano et al., 2012). Socioeconomic and cultural determinants affect drug use, leading to marginalization and restricted access to healthcare (Khan, 2011; Farrington et al., 2012; Carpenter, 2015).

The increasing frequency of drug use, especially among the youth in Pakistan, is a substantial health risk to the country. Over 6.7 million individuals in Pakistan are involved in drug use. The most common drugs are cannabis, heroin and amphetamines (Farooq et al., 2017; Zada et al., 2022). Approximately 25% of teenagers consistently use cannabis, while growing tendencies such as crystal methamphetamine and sheesha smoking are prevalent in metropolitan areas and among middle-class demographics (Masood and Us Sahar, 2014; Atif et al., 2020). Unmanaged peer pressure, financial and mental health issues increase the likelihood of young individuals becoming drug addicts (Ahmed et al., 2022). Cultural restrictions and gender norms in conservative regions impede drug use prevention and treatment efforts, requiring culturally tailored treatments for young women (van Solinge, 2007; Khan et al., 2024).

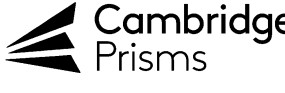



According to Khan et al. (2017), over a quarter of Pakistani teenagers are unaware of the harmful consequences of psychoactive drugs. The commencement of drug use among young individuals is influenced by several variables, including ignorance, social pressure, cultural expectations, familial relationships and feelings of isolation (Wolfe et al., 2008). The use of legal tobacco and alcohol often normalizes drug use among young individuals, with a notable correlation between alcohol intake and chronic illnesses, mental health issues and familial discord (Rose and Cherpitel, 2011). These patterns show that a mix of social and knowledge gaps makes young individuals more likely to engage in dangerous behavior. In this regard, the public's education and problem-avoidance skills are of utmost importance (Khan et al., 2025).

The increase in drug availability in Pakistan, particularly in Khyber Pakhtunkhwa, can be attributed to the nation's closeness to Afghanistan, which is recognized as the top opium producer globally (Ward and Byrd, 2004; Malik and Sarfaraz, 2011). Pakistan exhibits one of the highest national rates of drug consumption globally, with more than 6.7 million individuals engaging in drug use (Zubair, 2015; Zada et al., 2022). Earlier investigations (Ali, 1999) demonstrated a typical pattern of cannabis usage, which persists in affecting the ongoing issue. It is alarming that nearly 700 individuals lose their lives each day in Pakistan due to drug-related problems (Dawn News, 2015). The report emphasizes the gravity of the drug issue and highlights the necessity of conducting studies within the community to develop targeted strategies for treatment and prevention.

### Factors influencing youth substance use

The mental health of young individuals, their cognitive growth, social connections, familial relationships and socioeconomic status all contribute to their patterns of drug use (Preston et al., 2021). Drug use often escalates as a response to stress and emotional turmoil, influenced by socioeconomic factors like poverty, financial insecurity and inadequate education (Kandel et al., 1978; Ghate and Hazel, 2002). Substance use serves as a means of coping for individuals facing poverty due to the limited social and educational opportunities available. The prevalence of drug use is heightened among individuals who experience inadequate emotional support from carers or come from families with a history of substance use (Farrell and White, 1998). Kilpatrick et al. (2000) indicate that young individuals often engage in substance use due to dysfunctional families, which are characterized by ineffective communication, parental discord and inadequate oversight.

Peer influence, family relationships and economic circumstances significantly shape the trends of substance use among young individuals (Marshall and Werb, 2010). Substance use and various risky behaviors are often encouraged within peer groups among young individuals to achieve social acceptance and a sense of belonging (Prinstein et al., 2001). Many young individuals resort to substance use like alcohol and narcotics to manage stress, despair, anxiety and other mental health issues (Wills, 1990; Khantzian and Albanese, 2008). The prefrontal cortex is crucial in young individuals for regulating impulses and making decisions. Individuals in development often engage in hazardous activities like substance use due to a lack of understanding of their consequences (Evans and Stanovich, 2013; Giedd, 2015). Various factors influence the rise in drug usage, necessitating the use of targeted therapies to tackle these challenges (Sinha, 2001).

## Methods

The study investigates the factors influencing drug use among young women in Khyber Pakhtunkhwa, Pakistan, focusing on social, psychological and economic factors. It aims to understand how these factors affect their drug use and contribute to their addiction.

### Research questions

We pursued the following research questions: (1) How do social factors influence young women's drug use? (2) How do psychological factors lead to drug use among young women? (3) How do economic factors contribute to drug addiction among young women?

### Research design

The study utilized a qualitative research design. We recruited participants through snowball sampling. Data were collected from 12 in-depth qualitative interviews conducted in the cities of Mardan and Peshawar between March and June 2022.

### Sampling and participant selection

Eligibility criteria included (1) identifying as a woman, (2) being between 18 and 21 years of age, (3) having a history of drug use (current, in recovery or ceased) and (4) demonstrating willingness and ability to share personal experiences in an interview setting. The evaluation of the screening process involved posing targeted questions to prospective participants, including inquiries like, "Are you at ease discussing your personal experiences with drug use?" And "Do you believe you are ready to speak candidly about the challenges you've encountered concerning substance use?"

The enthusiasm shown by participants in the study indicated their preparedness, as they expressed a desire to engage in the interview process. Participants were required to uphold both

**Table 1.** Demographic and drug use status of participants

| Participant characteristics | Number of participants (*N* = 12) |
|---|---|
| Age range | 18–21 years |
| **Current drug use status** | |
| Ceased drug use | 2 |
| Under recovery treatment | 4 |
| Currently using drugs | 6 |
| **Education level** | |
| Completed primary school | 4 |
| Completed middle school | 6 |
| Illiterate | 2 |
| **Employment status** | |
| Students | 4 |
| Unemployed | 7 |
| Employed | 1 |
| Residency | Mardan and Peshawar, Khyber Pakhtunkhwa, Pakistan |

mental and physical stability to engage in the interview without facing immediate distress. At each stage of the interview, we assessed the participants to determine their readiness to proceed. We prompted the interview participants to share their experiences of feeling overwhelmed, anxious or physically uncomfortable. We provided the opportunity for individuals exhibiting signs of distress, such as panic attacks, severe anxiety or physical discomfort, to pause the interview and take a break. We ended the interview and removed the participant from the study to prioritize their well-being if they showed signs of distress. Participants had to reside in Mardan or Peshawar, and all participants were required to provide informed consent before participating in the study. Participants were required to be prepared to share their experiences and maintain mental and physical stability to avoid distress during an interview.

The original intention for the final sample size did not include getting 12 people to participate. The sample size was determined continually, according to the concept of data saturation. We continued to hire people until we reached saturation, and after each session, we reviewed the interviews. Data gathering stopped when no new themes or important insights came up. The research team assessed theoretical saturation throughout the data collection phase by analyzing transcripts from each interview and evaluating the feasibility of integrating further material. The method included evaluating both recognized topics, such as socioeconomic variables influencing drug use, and emergent themes that arose during interviews, including emotional coping techniques and social support systems. Recruiting participants for the study proved challenging due to the sensitive nature of the issue, forcing the researchers to rely on snowball sampling through local networks. The study drew in 12 participants, thereby presenting a diverse range of perspectives.

We identified our first participants by networking with individuals we knew who worked in addiction treatment, community health and social work. The study involved rehabilitation centers, local non-governmental organizations (NGOs) and community health clinics that assist individuals dealing with drug use issues. We examined the institutions' direct engagement with individuals struggling with addiction and the effectiveness of their treatment programs. These organizations and our team successfully collaborated due to our professional connections, past partnerships and discussions with specialists in the field. Social workers and healthcare professionals identified individuals who met the criteria and expressed a willingness to take part in the study. In this study, the snowball sampling method was employed. Scholars may connect with marginalized communities and gather a rich array of qualitative data through the snowball sampling method (Dusek et al., 2015). Employing snowball sampling methods tends to increase individuals' engagement in social networks (Cohen and Arieli, 2011).

(Table 1): A sample of 12 participants ($n = 12$) was available during the interview. Seven participants ($n = 7$) were interviewed in Mardan, while five ($n = 5$) were interviewed in Peshawar. Due to participant availability and accessibility, Mardan had a slightly higher number of participants, which was attributed to stronger connections with local healthcare practitioners and social workers who assisted in the recruitment process. All the participants were women. The participants varied in age, ranging from 18 to 21 years. Participants' educational backgrounds varied, with some having completed primary or middle school and others being illiterate. This pattern reflects national trends, with female literacy rates lower in Pakistan, especially in Khyber

**Table 2.** Types of substances used by participants ($N = 12$)

| Substance | Number of participants ($N$) |
|---|---|
| Opioids | 4 |
| Crystal methamphetamine | 6 |
| Cannabis | 5 |
| Heroin | 2 |

Pakhtunkhwa. Out of the total, four participants ($n = 4$) had completed primary school, six ($n = 6$) had finished middle school and two ($n = 2$) were illiterate. Four participants ($n = 4$) were students, seven ($n = 7$) were unemployed and one ($n = 1$) had a job. Most participants came from low socioeconomic backgrounds, characterized by financial instability, limited educational opportunities and high unemployment rates.

Table 2: During the interviews, two individuals ($n = 2$) reported that they had ceased drug use; one had maintained abstinence for 6 months, while the other had done so for nearly a year. Both individuals have a history of using cannabis and crystal methamphetamine, with one also having previously used opioids. Four ($n = 4$) were currently undergoing a recovery treatment program at community-based centers, where they received counseling, medical support and behavioral therapy for detoxification, relapse prevention and psychological counseling. Meanwhile, six participants ($n = 6$) were still using drugs. Individuals reported using various substances, including cannabis, crystal methamphetamine and heroin, and often engaged in the simultaneous use of multiple substances, influenced by factors such as availability, peer pressure and personal coping strategies. The individuals involved were residents of Khyber Pakhtunkhwa, particularly from Mardan and Peshawar, two cities located in Pakistan.

### The interview protocols

After explaining the nature of the research project to potential participants and undergoing a thorough ethical review process that emphasized the sensitivity of the data and the sample size, we successfully obtained the required approvals from the study participants in the cities of Mardan and Peshawar. Using snowball sampling, we identified more potential participants who met the study's data collection criteria through referrals from our initial contacts. For this reason, we conducted interviews with these individuals to verify that they were familiar with the project's rules and goals. We maintained high ethical standards by promising to keep their information private throughout the research. Given their safety and comfort for attendees, locations such as private schools and community centers were chosen. The intention behind selecting this location was to foster cross-cultural understanding and ensure that all participants felt at ease expressing themselves openly.

### Interview process

It is important to note that cultural norms and traditions in Khyber Pakhtunkhwa, combined with the segregated society, restrict male individuals from directly engaging with female participants. In this regard, female assistants, equipped with a profound understanding of qualitative analysis and data collection, helped carry out the interviews. We selected these female assistants due to their relationships with the participants and their insight into the cultural

context. Research assistants fluent in both English and Pashto ensured the accuracy of the translation and transcription of participants' work. Interviewers demonstrated exceptional competence in discussing sensitive topics such as substance use and gender dynamics in public health. The interview guides for the study were developed based on a thorough review of the existing literature. We consulted experts in gender studies, public health and psychology to ensure that questions were appropriate, culturally sensitive and reflective of the participants' experiences. Guides were first written in English and then translated into Pashto for improved comprehension and clarity.

We created a series of cross-questions to investigate the backgrounds of individuals who are struggling with substance use. Through open-ended interviews, we examined the social, economic and psychological factors contributing to drug addiction among young women. We investigated the participants' drug usage, behaviors and choices. The qualitative data collected through this method highlighted the various factors contributing to drug addiction in this context. We divided the interviews into four distinct sections. In the initial section, we inquired about factors such as age, educational attainment, family structure and living arrangements. The second section explored the various societal factors that influence drug use in young women. The third section discusses the intricacies of substance use and its underlying mental causes. The fourth section explored the field of economics.

Every 3-h interview was recorded with the participant's consent. We ensured the accuracy of the data by carefully transcribing the recordings and conducting a thorough probing. The questions were simple, which facilitated a more in-depth exploration of emotions and reflections. We conducted additional probes to expand our understanding of drug addiction and our questions. Before the interviews began, each participant reviewed a detailed consent form with the interviewer. Each participant voluntarily signed the consent form, confirming that they understood the objectives of the study and felt comfortable with their participation. Participants were assured of confidentiality to encourage open and honest discussion. Interviews were initially conducted in Pashto, the participants' native language, and then translated into English. No study participant received any specific compensation. Participants were thanked for their time and effort in the form of refreshments and reimbursement of travel expenses.

### Data analysis

We used a grounded-theoretic approach to social theory and social structure to examine the interview data, following Merton's (1968) suggestion, by searching for repeating themes. The interviews were transcribed verbatim by trained research assistants who were fluent in both Pashto and English. Transcriptions of the Pashto interviews were subsequently translated into English for research. Two researchers double-checked the transcripts against the original audio recordings to ensure their accuracy before completing the data for coding. Both individuals independently classified each transcript to ensure consistency and remove any potential for bias. The coding technique was followed by comparing codes and resolving any differences. A third senior member reviewed the transcript and attempted to resolve any disagreements among the team. We built a framework that efficiently structured all the data by using codes and categories.

We then employed inquiry questions, methodological design and field observations to create a thematic framework for the data. The interview transcripts were systematically coded in alignment

with the pre-identified themes. We undertook a thorough analysis and interpretation of the structured data, carefully categorizing variables including age, gender and occupation. A comprehensive study was conducted on the categories, featuring an in-depth review of the insightful quotes shared by the respondents to verify the authenticity of the interviews. Prominent themes that surfaced include the backgrounds of the participants, the impact of peer groups, challenges related to mental health and issues surrounding financial difficulties or unemployment. We engaged with the participants and examined elements influencing their substance dependence through the lens of social identity theory (Hogg, 2016), social learning theory (Bandura and Walters, 1977), cognitive dissonance theory (Morvan and O'Connor, 2017), self-medication theory (Khantzian, 1987) and strain theory (Agnew, 1992). We conduct a thorough analysis, literature review and theoretical frameworks to explore the experiences of young women in drug use.

### Ethics consideration

We ensured confidentiality of interview transcripts by assigning unique study IDs, removing personally identifiable information during transcription, securely storing data in a password-protected database and deleting audio recordings post-transcription, ensuring participants' responses were only used for research purposes.

### Findings

The findings reveal the influence of peer groups, mental health issues, financial difficulties or unemployment, followed by social, economic and psychological factors. Thematic analysis of interview data reveals the following findings, with several participants sharing their experiences.

### *The influence of peer groups*

Participants frequently emphasized the role of peers in initiating and sustaining substance use. Social gatherings, peer pressure and group activities often created pressure to experiment with drugs, particularly when refusal risked weakening relationships.

> "*During my first experience at a social gathering with a friend and her colleagues, I made the unwise decision to take ice (crystal meth) in an attempt to fit in with the prevailing atmosphere, as it seemed many others were behaving similarly.*" (Participant 2)

> "*My best friend started using different drugs and constantly offered it to me.*" After a few instances of saying "no," it became more challenging to deny without experiencing discomfort. I felt we wouldn't be as close if I didn't join." (Participant 5)

> "*My friends' seemingly pleasurable experiences under the influence of substances piqued my interest. Their laughter and carefree demeanor gave the experience an appealing look, making me curious about the effects firsthand. This curiosity sparked a desire within me to give it a try, to see if it could provide me with the same feeling of euphoria and escape that they appeared to be experiencing.*" (Participant 8)

> "*A friend of mine introduced me to vaping, which started my journey into addiction. We regularly participate in events together, and over time, the habit became increasingly crucial to both of us.*" (Participant 9)

> "*Our friendship began with minor issues and evolved into a complex relationship. As we became more daring, we experimented with harder drugs, heightened our connection to substance use, and complicated our relationship with risky behaviors.*" (Participant 11)

The narratives emphasize the significant role of peer pressure and social influence on drug use. Individuals often imitate their peers' behaviors to avoid rejection and maintain relationships. Social identity theory suggests that individuals may use drugs to conform to a group's identity, while cognitive dissonance theory suggests that individuals may change their attitudes.

### Mental health issues

Participants frequently described using substances to cope with anxiety, stress and emotional struggles. Drugs and medications were seen as temporary solutions that provided immediate relief, even if the effects were short-lived.

> "*I had a hard time coping with the pressure of school, especially the days before exams. It was a considerable sum. I started taking drugs that someone had given me at a party to help me deal with my anxiety and control my emotions.*" (Participant 3)

> "*Every day, the very act of thinking about the day ahead made me anxious. Smoking seemed to help ease that anxiety, if only for a few hours.*" (Participant 4)

> "*Whenever I experienced anxiety attacks, they were truly debilitating. Even though they were not prescribed to me, I found that some medications gave me immediate comfort. I just continued to rely on them to get through the day.*" (Participant 7)

> "*Coping with the daily demands, I found myself constantly on edge. A friend who found out about this offered me a joint. At first it was helpful, but soon I was smoking every day to feel normal.*" (Participant 8)

> "*My despair was only compounded by the loneliness I felt, which led me to turn to drugs as a means of short-term escape when I had no one else to turn to. They gave me a short-term boost to my mood and a sense of friendship, which helped to fill the emptiness I felt inside. Even though I knew it was harmful, I couldn't resist the short-term comfort they offered.*" (Participant 10)

> "*My mood swings were unpredictable; they would soar high one minute and crash the next. Even though I was aware of the potential consequences, I resorted to medication to restore balance. Because I tended to act impulsively, I regularly made hasty choices, and medication was no exception. It gave me an easy way out of overwhelming feelings, but I often regretted my decision when the consequences began to take their toll.*" (Participant 11)

The interviews reveal a rise in substance use as a coping mechanism for anxiety and stress, especially academic pressure, leading to a harmful addiction cycle. They point out that there must be interventions focusing on emotional regulation and impulse control to address underlying anxiety and stress.

### Financial difficulties or unemployment

Participants reported financial hardship, unemployment and unfulfilled ambitions as factors contributing to substance use, often presenting drugs as a temporary escape from poverty and job loss.

> "*After I was laid off, I began to feel depressed as my expenses continued to mount, leaving me feeling trapped. Even if it were only for a short period, I would turn to drugs to escape my money worries. Growing up in a time of constant scarcity, I found myself drawn to people who used drugs to find pleasure despite the hardships they were experiencing. Soon, I began using drugs myself, seeking a similar escape from the unrelenting weight of financial constraints.*" (Participant 4)

> "*I could not afford college and witnessed my friends depart for further education. My sadness and frustration at being abandoned and not*

> *achieving my goals drove me to drugs. The immense stress and instability I experienced at home due to both my parents losing their jobs during the recession further pushed me to use drugs as a means of coping.*" (Participant 8)

> "*My self-esteem plummeted after a year of unemployment, and I turned to drugs as my only source of happiness. Losing my job felt like losing part of myself, and to numb the pain of feeling worthless, I began using drugs more frequently. What started as an escape quickly became a daily necessity as my unemployment dragged on.*" (Participant 9)

> "*I experienced constant financial instability, reverting to old patterns after setbacks. Drugs became my coping mechanism, providing a temporary sense of control and relief from uncertainty.*" (Participant 12)

Narratives show individuals coping with emotional and financial challenges through substance use, resource reduction or a combination of both, emphasizing the need to address economic and social factors to reduce drug use.

### Discussions

This study illuminates the complex interactions of socioeconomic, psychological and social factors that affect drug use among young women in Khyber Pakhtunkhwa. Given that individuals frequently come across substances through their social circles and interactions, relationships with peers significantly influence the onset of drug use. Resistance can result in social marginalization or diminished social connections (Steinberg et al., 1994; Farrington et al., 2012). Reports from young women regarding changes in behavior aimed at alleviating social stress align with cognitive dissonance theory (Morvan and O'Connor, 2017) and social learning theory (Bandura and Walters, 1977), which suggests that behaviors are acquired through observation and imitation. Research by Henneberger et al. (2021), Piña et al. (2018) and Prinstein et al. (2001) shows that peer influence significantly predicts drug use among young individuals. This highlights the importance of preventive initiatives that incorporate peer groups, such as those educating on refusal skills and providing peer-led information.

Psychological distress has been acknowledged as a significant factor influencing drug use behaviors (Hawkins et al., 1992; Goldberg, 2012; Carpenter, 2015). Participants reported that they used drugs to manage stress, worry, schoolwork, loneliness and mood swings. These narratives align with the self-medication theory (Khantzian, 1987; Kaminer et al., 2011), which posits that individuals use drugs as an ineffective means to alleviate psychological pain (Wills, 2013; Seaward, 2017). Narratives elucidating impulsive behavior amid mood variations further substantiate the correlation between impulsivity and addiction risk (World Health Organization, 2004; Perry and Carroll, 2008; Smith and Cyders, 2016). These results underscore the need to integrate mental health services into both preventative and treatment programs, highlighting treatments that promote stress management, emotional regulation and impulse control. This discovery is essential in traditional contexts where mental health services are difficult to come by and negative ideas about mental health persist.

The prevalence of drug use was notably higher among individuals facing economic difficulties and unemployment. Participants indicated that they resorted to drugs as a temporary means to cope with financial instability, job loss, and unmet educational aspirations (Henkel, 2011; Khan, 2011; Frasquilho et al., 2015). These events exemplify a theoretical framework that suggests individuals

might resort to antisocial behaviors as a coping mechanism for the disappointment experienced when their aspirations and dreams are unfulfilled (Henkel, 2011; Frasquilho et al., 2015). This viewpoint aligns with stress theory (Agnew, 1992), which posits that economically disadvantaged cultures tend to depend more on maladaptive behaviors such as substance use due to a deficiency in effective coping strategies (Lazarus and Folkman, 1986). In Khyber Pakhtunkhwa, young women encounter significant economic challenges and gender-related barriers to both education and employment. As a result, some may resort to drug use as a means of coping or seeking empowerment.

Familial relationships, in addition to these features, significantly shape the emergence of susceptibility. Attachment theory, as proposed by Bowlby (1979), emphasizes the significance of early emotional bonds in shaping future behaviors. Research suggests that young individuals are more likely to use drugs due to parental substance use and dysfunctional interactions between parents and children (Farrell and White, 1998). The dual-process theory of decision-making (Evans and Stanovich, 2013) provides an understanding of the participants' impulsivity. Giedd (2015) notes that the immature prefrontal cortex in youngsters increases their propensity for risky behaviors, such as experimenting with drugs. When considered collectively, these concepts illustrate the multitude of factors that interplay, particularly the social, relational and developmental contexts of young girls and young women, which collectively shape drug use behaviors.

The investigation emphasizes the value of thorough and culturally aware strategies that address multiple facets of susceptibility. Programs should incorporate education that emphasizes peer interaction, provide mental health resources, promote economic empowerment efforts and create nurturing family settings to enhance communication and build resilience. Interventions must be customized to fit the cultural landscape of Khyber Pakhtunkhwa, where distinct challenges stem from patriarchal norms and limitations on women's mobility. By tackling social, psychological and economic stressors at the same time, prevention and treatment initiatives can more effectively assist young women, diminish dependence on substances as coping strategies and foster healthier, more resilient communities.

**Implications for research:** This study fills a significant knowledge gap and contributes to the growing body of qualitative research on women's drug use in conservative countries. Larger multisite studies across Pakistan are needed to examine regional differences, as the limited sample size of this study limits the generalizability of its findings. Longitudinal studies need to explore the impact of social pressures, mental illness and financial instability on drug use and recovery. Comparative studies in South Asia can shed light on cultural norms, gender expectations and socioeconomic constraints. It is critical to include the perspectives of communities, families and service providers in future research.

**Implications for policy:** The findings emphasize the necessity for drug policies that consider gender in Khyber Pakhtunkhwa, underscoring the susceptibility of young women. The investigation suggests ways to enhance mental health frameworks, promote women's economic empowerment via education, training and job initiatives, and integrate substance use prevention into youth educational curricula, while pointing out the importance of culturally attuned policies and community involvement.

**Implications for practice:** The research suggests a multidisciplinary approach for drug use therapy, involving mental health counseling, peer support networks, safe spaces for young women and job counseling and financial literacy training. Collaboration between NGOs, schools, healthcare professionals and local communities is crucial for prevention, treatment and long-term recovery to work.

## Conclusion

This study examines factors influencing substance use among young women in Khyber Pakhtunkhwa, Pakistan. The study finds that social influences, psychological stress and economic hardship play a critical role in substance use, which often leads to relapse. The findings suggest that substance use among young women in conservative settings occurs as a response to systemic and interpersonal challenges. The study emphasizes the importance of analyzing substance use from cultural, social and economic perspectives, while also addressing differences in contemporary beliefs about drug addiction among genders. An emphasis on peer-focused prevention, increased access to mental health services and initiatives to promote financial independence can achieve successful drug addiction treatment.

**Open peer review.** To view the open peer review materials for this article, please visit http://doi.org/10.1017/gmh.2025.10071.

**Data availability statement.** The study's data are unavailable to the public or inaccessible upon request, as this information could compromise the privacy of the research participants.

**Author contribution.** Muhammad Suhail Khan: Conceptualized the study and developed the methodology, conducting a literature review while creating the original draft, performing data analysis and drafting the manuscript. Yao Dewei and Wu Zongyou: Supervised the project. Aman Khan, Ibrahim, and Anastasiia Pavlova: Conducted interviews and contributed to interpreting the results and providing critical revisions.

**Financial support.** This research received no specific grant from any funding agency, commercial or not-for-profit sectors.

**Competing interests.** The authors declare none.

**Ethics standard.** The project was approved by the Ethics Committee of the School of Sociology and Political Science of Anhui University on May 20, 2021. Interviewed participants provided written informed consent with their signatures.

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
