## [Reviewer Report]

The study titled “Drug abuse among adolescent girls: a qualitative study of socio-economic and psychological perspectives” provides important insights into the issue of drug addiction among young girls in Khyber Pakhtunkhwa, Pakistan. The research employs comprehensive interviews and theoretical frameworks such as social identity theory and the self-medication hypothesis to provide context for the findings.

Nevertheless, the sample size of 12 participants may allow for more applicability of the findings. The paper would be enhanced by incorporating more recent studies on adolescent drug use, especially those that concentrate on the South Asian region or provide comparative analyses with other developing countries. An in-depth exploration of societal norms and gender dynamics in the area may yield valuable insights. The paper may also propose more comprehensive, evidence-driven intervention strategies, including mental health services, peer support, and economic empowerment programs.

---

## [Reviewer Report]

Dear Author,

Drug abuse in Pakistan is a very serious issue. Educated and even the educated youth are trapped in this menace. I have thoroughly checked your article. I have found some mistakes which if incorporated will shine your research work. Following are a few suggestions;

Revisit your reference, Journal name and issue numbers are missing.

Your methodology is not clear please elaborate on it further

Replace the word “data was‘ with ’data were”

Thanks

---

## [Reviewer Report]

This paper presents qualitative findings from in-depth interviews with adolescent girls (i.e., women aged 18 to 21) in Khyber Pakhtunkhwa, Pakistan, focused on investigating socioeconomic and psychological factors that influence drug use. It is clear that the authors thoroughly researched theories surrounding substance use in young adults, and that much thought went into the design and execution of this study. While the themes identified in the interviews themselves are not unique (i.e., using substances to fit in with peers, substance use as a coping mechanism), this study is among the first to assess reasons for substance use among young women in Pakistan. Given the high burden of substance use in Pakistan, and the lack of research on substance use among women in this country, findings from the present study do start to fill a gap in the literature.

Despite this, I have some major concerns with how the manuscript is written, and how the study was conducted, which I have outlined below. I strongly recommend that the authors address the following comments before the manuscript is reconsidered for publication.

<b>Major Comments:</b>

1. This journal requires that research involving human participants be approved by appropriate institutional ethics committee(s) and conform to relevant international ethical and legal standards for research. While the authors state that “informed consent was obtained” and “confidentiality was strictly maintained,” they do not mention obtaining approval from an ethics board. I strongly suggest that the authors include the name of ethics board(s) and protocol number(s) in the Ethics Statement. I also recommend that the authors add a statement about receiving ethics approval in the main body of the text. If this research was not approved by an ethics board, this should also be clearly stated, along with justification for why an ethics board was not used.

2. The authors spend over thirteen pages (more than half of the manuscript) providing the reader with background information on substance use in Pakistan, substance use among adolescents, and negative effects of substance use. While this information is interesting, I do not believe this level of detail is needed to understand the study. In fact, I personally found that this level of detail distracted from the actual study. In contrast, I think the manuscript would be strengthened if more detail were added to the Methods section (detailed below). Further, more information should be added to the Discussion section, which is currently a single paragraph. Finally, given that the manuscript is over 3000 words over the research article word limit of 5000 words, I recommend that the authors make the following changes to text’s structure:

a. The Introduction should be significantly shortened, focusing on just the background details most pertinent to the current study. To ensure they are getting their main points across, I also recommend that the researchers summarize existing research, rather than detailing it. Likewise, I recommend removing the two maps included in the Introduction (while interesting, they are both over a decade old, and do not directly relate to the present study). Finally, the authors may choose to move some of the information on theories of adolescent drug use to the Discussion section, where they can more clearly integrate their findings into the existing literature.

b. I recommend that the authors add the following information to the Methods section:

i. Eligibility criteria for participants

ii. How/why a sample size of 12 participants was chosen

iii. How initial participants were identified. While the authors say they were “identified with the help of healthcare providers and social workers,” it is unclear to me how the researchers contacted these healthcare providers/social workers. Did the researchers already have relationships with certain healthcare facilities? If so, which types of facilities? How were these facilities chosen?

iv. Information on the “female colleagues” who conducted the interviews – were these research assistants, professors, etc.? Were they bilingual, or did they only speak Pashto? Did they have previous experience conducting similar interviews?

v. Processes for developing interview guides

vi. Whether patients were compensated for interviews (and if so, how much)

vii. Substances used by participants

viii. How long the two participants had stopped using substances, and which types of substances

ix. Further define the “recovery treatment program” that four participants were attending

x. How many participants were interviewed in Mardan vs. Peshawar

xi. How the participant sample compares to the overall population – for instance, is the level of education typical for women in Pakistan or atypical? In general, were participants of high, middle, or low socioeconomic status?

xii. How were interviews transcribed / who transcribed them?

xiii. What steps were taken to ensure confidentiality – for instance, were transcripts deidentified with study IDs? Where was data stored?

xiv. Who was involved in the coding process? How many coders coded each transcript? How were discrepancies between coders resolved?

c. The authors should expand on the Discussion section to better contextualize their findings within the existing literature and discuss strengths and limitations of the study. The authors may also want to comment on the rationale for separating the themes of “mental health issues,” and “financial difficulties or unemployment,” as both themes seem to point to using substances to cope with difficulties.

3. The objectives of this study were to explore factors influencing drug use among adolescent girls in Pakistan. Yet, the study only enrolled young adults aged 18 or older. The World Health Organization currently defines adolescence as 10 to 19 years of age. Given the discrepancy between this definition and the age of participants, I suggest that the authors either (a) consider describing their population as “young women” instead of adolescent girls or (b) provide rationale as to why women aged 18 – 21 are considered adolescents in Pakistan.

4. The authors use language like “drug abuse” and “drug users” throughout the manuscript. Recently, many researchers and institutions in the US and other countries have started moving away from this language due to its potentially stigmatizing nature. I recommend substituting “drug abuse” with “drug use,” and using person-first language when referring to people who use drugs. When considering language, the authors may find the following resources helpful:

https://nida.nih.gov/nidamed-medical-health-professionals/health-professions-education/words-matter-terms-to-use-avoid-when-talking-about-addiction

https://www.canada.ca/en/health-canada/services/publications/healthy-living/stigma-why-words-matter-fact-sheet.html

https://www.recoveryanswers.org/addiction-ary/

<b>Minor Comments:</b>

5. I recommend that the authors double check the formatting of this manuscript against this journal’s guidelines. I believe this manuscript is currently over the world limit and may not be using the recommended citation style. Additionally, the authors should include the Impact Statement above the Introduction, rather than within the Introduction, as this was initially unclear.

6. The authors state that details of the research funding are currently restricted by supervisors. I recommend the authors double check with the editor to ensure that the journal allows this restriction of information.

7. The authors state that “cultural norms and traditions, along with the segregated nature of the study region, have rendered the data supporting the study’s findings unavailable.” Based on this statement alone, it is unclear to me how norms/traditions/ the segregated nature of the study region precludes de-identified transcripts or a codebook from being made available upon request. The authors may want to elaborate upon this in the Methods section.

---

## [Editor Report]

Thank you for submitting your manuscript to our journal. We feel your manuscript provides an important addition to the literature, but some major revisions are required as indicated by the third reviewer. Please also consider adding reference to Pakistan in your title as this study was specific to this population. We look forward to receiving a revised manuscript.

---

## [Reviewer Report]

This paper presents qualitative findings from in-depth interviews with adolescent girls (i.e., women aged 18 to 21) in Khyber Pakhtunkhwa, Pakistan, focused on investigating socioeconomic and psychological factors that influence drug use. The authors were responsive to most of the feedback in the first round of reviews, resulting in a much stronger manuscript. While the majority of comments were addressed, not all comments were addressed, and I still have some lingering questions/suggestions for the authors. I suggest that these comments are addressed before the manuscript is accepted for publication.

1. Eligibility Criteria: The authors added much more information to the eligibility criteria, thus strengthening the methods section. However, I still have some clarifying questions:

1a. Eligibility criteria included that participants “demonstrate a readiness to share their experiences.” How was this measured? Were specific questions asked during a screening process, or was interest in the study considered to demonstrate readiness?

1b. Eligibility criteria included that participants “maintain mental and physical stability to ensure they can engage in an interview without experiencing immediate distress.” How was this measured? Or, were there specific criteria around when and how participants were to be excluded (for instance, if they displayed certain symptoms during the interviews)?

1c. The sentence “individuals residing in Khyber Pakhtunkhwa, particularly in Mardan and Peshawar, must give informed consent” makes it sound like only participants in these two locations had to give informed consent. I think this sentence would be clearer if you rephrased it to say something like “participants had to reside in Mardan or Peshawar, and all participants had to give informed consent prior to beginning the study.”

2. Sample Size: The authors state that “based on data saturation, we decided on a sample size of 12 people…”. Does this mean that themes were assessed on an ongoing basis and recruitment stopped once saturation was reached? Or that 12 was the anticipated number of participants needed to reach theoretical saturation (if so, please cite previous research suggesting 12 participants may be sufficient)? Or were only 12 people available because this is a hard-to-reach population? The authors should clarify (1) whether recruiting 12 people was the initial plan or not; (2) whether theoretical saturation was assessed on an ongoing basis or not; and (3) if assessed on an ongoing basis, if this included just primary pre-determined themes, or secondary themes as well.

3. In my previous review, I wrote “The authors may also want to comment on the rationale for separating the themes of “mental health issues” and “financial difficulties or unemployment,” as both themes seem to point to using substances to cope with difficulties.” While the authors now do talk about mental health and financial difficulties as two separate factors contributing to drug use in the discussion section, they do not directly discuss why they chose to consider these two different themes. Please clarify why you chose to separate these two themes, although they are both themes about coping. The discussion section would be an appropriate place to comment upon this.

4. (Not addressed from previous review): In my previous review, I wrote:

“The objectives of this study were to explore factors influencing drug use among adolescent girls in Pakistan. Yet, the study only enrolled young adults aged 18 or older. The World Health Organization currently defines adolescence as 10 to 19 years of age. Given the discrepancy between this definition and the age of participants, I suggest that the authors either (a) consider describing their population as “young women” instead of adolescent girls or (b) provide rationale as to why women aged 18 – 21 are considered adolescents in Pakistan.”

This comment was not addressed in the previous response to reviewers. Kindly address this comment.

5. Regarding potentially stigmatizing language, the authors were responsive and changed the term “drug abuse” to “drug use” throughout the manuscript. However, while the authors state in their response that they changed “drug users” to “people who use drugs,” I could not find any evidence of this in the manuscript. I recommend that the authors change “drug users” to person-first language throughout the manuscript.

6. The authors were very responsive to cutting text. However, I believe the manuscript is still ~500 words over the journal’s word limit. While I do not personally have a problem with the longer manuscript (and default to the editor’s opinion on this manner), I think there are many places in the introduction where the authors could significantly shorten their text. Specifically, I recommend that the authors:

6a. Summarize versus list data: For instance, in the first paragraph, the authors may consider removing specific statistics on who uses drugs worldwide and drug types, and instead say something like “drug use increases the risk of HIV, hepatitis C, and other health complications, thus increasing its public health significance.” Likewise, in paragraph 2, the authors could just say “commonly used substances in Pakistan include…” rather than using specific statistics. Changes like this can be made throughout the introduction.

6b. Turn the section titled “Factors contributing to adolescent substance use” into just one or two paragraphs: There is a lot of information on various factors that may contribute to adolescent substance use. While this is interesting, I do not think this level of information is needed. I think it is sufficient to say something like “In other populations, factors like socioeconomic status, family dynamics, peer influence, mental health, and cognitive development have been shown to contribute to adolescent substance use. For instance…”. The authors can also move some of this detail to their discussion (as the discussion of findings is still very brief).

Minor Comments:

7. The authors did a good job providing more information on the people who conducted the interviews. I recommend clarifying that they were research assistants, rather than just calling them “skilled assistants.”

8. The authors did a good job providing information on the types of substances that were used by participants. I recommend adding this information to the table (e.g., Opioids = N, Methamphetamine = N, Cannabis = M, etc.)

---

## [Editor Report]

Thank you for the revisions to your manuscript which, as stated by the reviewer, have greatly strengthened your manuscript. Please note, however, that some additional minor revisions are required before the manuscript can be published. We appreciate your continued efforts and look forward to receiving your revised draft.

---

## [Reviewer Report]

This paper presents qualitative findings from in-depth interviews with women aged 18 to 21 in Khyber Pakhtunkhwa, Pakistan, focused on investigating socioeconomic and psychological factors that influence drug use. This is my third time reviewing this manuscript. The authors were very responsive to feedback and adequately addressed all previously stated concerns.

While I do not have concerns with the science, and believe this manuscript is relevant for this journal, I still have some concerns with the clarity of the writing and organization of the paper (detailed below). It is possible that these concerns are based on my own stylistic preferences/ that the journal does not share these views. Therefore, I defer to the editor on whether these concerns should be addressed before publication, or whether the manuscript is suitable for publication as it is currently written.

While I am formally recommending a minor revision to convey my concerns with the clarity of the writing, based on the science alone, I do recommend this manuscript for publication.

Clarity of writing:

- I personally believe that the introduction would be strengthened if the authors shortened it and made it more focused. Some specific suggestions are:

o Combine paragraphs 1 & 2, or make paragraph 1 focused on substance use in general and paragraph 2 focused on substance use specifically in Pakistan. I found going back and forth between the world and Pakistan somewhat confusing.

o Consider removing mention of substances like caffeine, unless there is a reason the authors want to focus on caffeine (and if so, this should be clarified).

o Ensure each paragraph in the introduction has a distinct and meaningful message. Right now, the paragraphs somewhat read like a list of facts. I think it would be much more clear to the reader if you summarized the takeaways of most of these facts, rather than stating them.

- Throughout the manuscript, there were times I felt like sentences were written awkwardly and I had difficulty fully grasping what the authors were trying to convey. For example (please note this is not an exhaustive list):

o Introduction: “The majority use heroin, with 29.5% developing cannabis addiction (Ali, 1999) - To me, this reads as if the authors are suggesting that people develop a cannabis addiction because they use heroin. I recommend rephrasing this sentence or removing it all together (as I am not sure what it adds to the introduction, and it is likely outdated).

o Objective: “How does the social factor influence young women’s drug use?” - I recommend rephrasing this as “how do social factors influence…” because right now it sounds like there is only one social factor.

o Research Design: “We collected the data using snowball sampling from 12 in-depth qualitative interviews…” - This reads as the sampling came from 12 interviews, rather than recruiting the 12 people for interviews via snowball sampling. I recommend placing a period after “snowball sampling” and making this two separate sentences.

o Sampling: “The study’s eligibility criteria stipulated that participants, including young women aged 18 to 21, must meet specific conditions.” - This sounds like there were also participants who were not 18 to 21. I recommend rephrasing this to say something like “Eligibility criteria included: (1) identifying as a woman; (2) aged 18 – 21; (3) possess a background of drug use…, etc.”

o Strengths and limitations: “However, the limited sample size may not fully capture the experiences of marginalized or less visible users. The results provide significant results.” - I would elaborate on how the study provides significant results, as it is confusing to have this conclusion immediately following a limitation.

Manuscript Organization:

- I do not think you need to put “Source: Authors’ analysis” under your tables. As you are presenting data from the study described in this manuscript, it is assumed that it comes from your analysis of this data.

- The Findings section seems to be a combination of a traditional results and discussion section, as the authors discuss findings as they present them. Given that there is a separate discussion section, I found this a little confusing. However, I ultimately defer to the journal on if they are okay with this manuscript organization.

---

## [Editor Report]

This manuscript has been greatly improved upon during the revision process and we look forward to publishing it in our journal. At this stage, we feel that the manuscript requires some slight improvements in the clarity of the writing prior to publication. The reviewer has provided some very good examples of ways to improve the clarity. We recommend that the authors edit the manuscript accordingly and do a thorough read through and polish to make it ready for publication. We look forward to receiving your revised draft.

---

## [Reviewer Report]

The authors thoroughly addressed all previous concerns. Procedures and findings are presented clearly and well-integrated into the existing literature in the discussion. I recommend this manuscript for publication.

---

## [Editor Report]

We are grateful for your ongoing revisions and commitment to publishing in this journal. Your manuscript has been greatly improved upon and we feel it is ready for publication. One final note is that some of the sentences appear to be highlighted in turquoise, but this will hopefully be addressed in the publication process. Thank you again for your dedication and hard work.